# LEARNING IMAGE REPRESENTATION FOR LIMITED CRYO-EM DATA

**Radosław Kuczbański**[1*]**, Wojciech Kozłowski**[1]**, Lukas Frey**[2]**,**
**Konrad Karanowski**[1]**, Mateusz Grzesiuk**[1]**, Andrzej Rzepiela**[2]**, Maciej Zięba**[1,3]
[1]WUST, Poland     [2]ETH Zurich, Switzerland     [3]Tooploox, Poland
*radoslaw.kuczbanski@pwr.edu.pl

## ABSTRACT

Cryo-electron microscopy (cryo-EM) enables the visualization of proteins in distinct structural states, but extracting robust information from experimental images remains challenging. In particular, learning useful representations of 2D class averages is limited by the scarcity of annotated real datasets, which constrains both model training and benchmarking. To address this gap, we introduce the image retrieval task for cryo-EM 2D class averages and propose a two-stage domain-mixed training paradigm. In the first stage, the model is pretrained on easily accessible synthetic 2D class averages to establish feature representations. In the second stage, it is finetuned on a small mixed synthetic-real dataset to adapt to experimental variability. We demonstrate that this approach enables effective retrieval under limited and imbalanced data conditions, significantly outperforming models trained only on real images. Our work establishes a scalable framework for bridging synthetic and experimental data in cryo-EM, with the potential to accelerate downstream structural analysis, such as protein identification.

## 1 INTRODUCTION

Understanding the structural and functional states of proteins is central to elucidating cellular mechanisms. Cells contain complex protein mixtures, and methods capable of analyzing multiple proteins simultaneously could greatly accelerate cryo-EM processing pipelines. Ultimately, developing approaches to screen whole-cell lysates, potentially capturing the most common proteins, would enable comprehensive monitoring of proteomic changes. Such strategies could, for example, reveal how drug treatments modify protein states across the entire cellular proteome.

Despite the biological importance of capturing protein conformations and proteomic changes, the computational analysis of experimental data remains challenging. In particular, learning robust 2D class average representations from experimental images is a difficult problem, largely because publicly available datasets consist primarily of raw, unprocessed data and lack annotated 2D class averages for such tasks. The limited availability of annotated real data limits the ability to train and benchmark computational models, forcing reliance on synthetic data or highly specialized datasets that may not generalize to real-world scenarios. Moreover, only a fraction of the human proteome has been structurally resolved by cryo-EM, with many proteins instead represented by X-ray crystallography models or synthetic structural data generated using methods such as AlphaFold. Overcoming these limitations is essential for developing scalable approaches that are capable of interpreting complex biological mixtures directly from experimental imaging data.

To address this challenge, we propose a two-stage training paradigm. In the first stage, models are pretrained on synthetic 2D class averages, which are readily generated and annotated. This pretraining allows the model to learn a robust representation of image features relevant to classification. In the second stage, the pretrained model is finetuned on a small mixed dataset, enabling adaptation to the specific characteristics of experimental data. This approach leverages the abundance of synthetic data while requiring only minimal real data for finetuning, and we demonstrate that it significantly outperforms training only on the limited real dataset.

The main contributions of our work can be summarized as follows:

- We introduce cross-domain image retrieval task with limited real data in the context of cryo-EM 2D class averages, and to the best of our knowledge, this is the first study to do so.

- We show that, even with a limited and highly imbalanced dataset, the task remains feasible when using class averages, enabled by a two-stage, domain-mixed training paradigm.

## 2 RELATED WORKS

**Cryo-electron microscopy (cryo-EM)** has become a powerful method for resolving macromolecular structures Kühlbrandt (2014); Cheng (2015). cryo-EM data processing involves several stages, including motion correction, CTF estimation, particle picking, 2D classification, and 3D refinement.

A key step is **2D class averaging**, which groups similar particle images to enhance the signal-to-noise ratio and reveal structural features. This is typically done using iterative alignment and clustering algorithms van Heel (1984); Sigworth (1998); Scheres (2012). 2D classification helps to filter out poor-quality particles, assess structural heterogeneity, and provide orientation estimates for 3D reconstruction. Advances in this area have improved speed and accuracy, making it an essential part of modern cryo-EM workflows.

**Class ranker** Kimanius et al. (2021) is a part of the data processing pipeline included in RELION Scheres (2012) that is designed to automate the selection of 2D class averages during cryo-EM data processing. The model consists of two main components: a CNN that extracts features from 2D class averages, and an MLP that integrates these features with additional metadata, such as alignment statistics and class size, to assign a quality score between zero and one. This score enables reproducible ranking of 2D class averages for downstream processing, reducing the need for manual selection. It should be underlined that this model only predicts the quality of a 2D class average and does not classify it.

**Cryo-IEF** Yan et al. (2024) is a deep learning foundation model for cryo-EM image evaluation that operates directly on noisy single-particle images rather than on 2D class averages. Pretrained on a large dataset of approximately 65 million particle images using unsupervised learning, Cryo-IEF has demonstrated strong performance across multiple tasks, including particle classification, pose clustering, and quality assessment. In addition, the CryoWizard pipeline, introduced with Cryo-IEF, combines the model with streamlined automation tools to deliver an end-to-end cryo-EM data processing solution, enabling more efficient and accessible structural biology workflows.

**CryoSPARC's Reference Based Auto Select 2D** Punjani et al. (2017) is an automated approach for selecting 2D class averages during cryo-EM processing integrated into CryoSPARC data processing suite. It requires a single 3D density map and works by generating all possible projections of that map and calculating their correlations to each 2D class average. Classes are then selected based on these correlation scores, enabling broader and more systematic class selection without manual intervention. This method does not analyze the content of the class averages themselves and cannot identify the specific protein present, relying entirely on the provided reference.

**Metric learning** aims to learn embedding functions so that the distance between samples reflects their semantic similarity. Early work Goldberger et al. (2004) proposed a method that optimizes a stochastic leave-one-out k-NN objective. Subsequently, Chopra et al. (2005) introduced the convolutional Siamese network for face verification. This was further generalized by Hadsell et al. (2006) by formulating the contrastive loss. Later, Hoffer & Ailon (2015) proposed the triplet network, which simultaneously considers positive and negative samples. This approach was scaled up in Schroff et al. (2015) by applying several mining strategies for more effective triplet selection. To further improve training stability, Wen et al. (2016); Movshovitz-Attias et al. (2017) incorporated class centers into the loss design.

# 3 DATASETS

In this section, we describe how we prepared both synthetic and real datasets to solve the task of learning image representation for cryo-EM 2D class averages data.

## 3.1 SYNTHETIC DATA

**Selecting proteins**  As a first step in preparing the synthetic dataset, we searched the RCSB Protein Data Bank Berman (2000) to obtain a curated set of macromolecular structures. To ensure that the selected entries were both biologically relevant and technically suitable for synthetic projection generation, we applied the following filtering criteria: (i) only structures determined by electron microscopy (EM) were considered; (ii) the reported resolution was required to be better than 4 Å; (iii) the molecular weight of the complex had to fall within the range of 100–1000 kDa; (iv) each entry was required to contain exactly one deposited polymer entity instance; and (v) entries containing RNA polymer entities were excluded.

The application of these filters yielded an initial set of 650 candidate structures. To minimize redundancy and avoid bias in the synthetic dataset, we subsequently queried the RCSB for structural neighbors of each candidate and discarded all structures exhibiting a similarity score above 0.5 to another candidate. This reduced the dataset to 186 unique entries.

Due to technical limitations in handling extremely large maps, a small subset of structures had to be excluded, resulting in a final working set of 175 entries. For each of these, the corresponding experimental density map deposited alongside the atomic model was retrieved and used as the starting point for synthetic dataset generation.

**Synthetic 2D class averages**  To generate synthetic 2D class averages, we produced 1000 random projections for each selected density map. This number was chosen to ensure sufficient sampling of the protein over the full range of viewing directions. Then, instead of running computationally expensive 2D classification, each projection was low-pass filtered during augmentation. This choice is motivated by the observation that experimental 2D class averages often resemble low-pass filtered projections of the underlying model Scheres (2012).

## 3.2 REAL DATA

For the real dataset, we used CryoPPP Dhakal et al. (2023), which aggregates cryo-EM data from 34 EMPIAR (Iudin et al. (2022)) entries. To ensure unambiguous structural correspondence, we excluded all EMPIAR entries linked to more than one published model entry, leaving us with 17 candidates. Each EMPIAR entry was then processed using RELION Scheres (2012) 2D classification.

Due to variability in data quality, only 10 entries produced usable class averages. Then, to select good class averages, we applied the RELION class ranker Kimanius et al. (2021) with a threshold of 0.3 across all classification iterations. The resulting set constitutes the real dataset, summarized in Table 1 and presented in Figure 1.

A key source of discrepancy between synthetic and real 2D class averages (as seen in Figure 1) comes from differences in data provenance within the cryo-EM processing pipeline. Synthetic samples are generated from real, published 3D density maps, which, at the final stage of processing, have undergone postprocessing, including masking and B-factor sharpening. In contrast, real 2D class averages originate from an earlier stage of the cryo-EM processing pipeline, before such refinements are applied. As a result, they typically exhibit lower signal-to-noise ratios, less well-defined structural features, and prominent lipid densities, particularly in the case of membrane proteins. This difference in processing stage leads to systematic appearance discrepancies that contribute to the domain gap between synthetic and experimental data.

Table 1: Class counts of real cryo-EM 2D class images, sorted in ascending order. The dataset is imbalanced, with as few as 20 samples in the rarest class and up to 601 in the most frequent class.

| Protein | 7knu | 7n9z | 6ujb | 5irx | 6wvj | 5u6o | 6kff | 6kfh | 7k61 | 5vkq |
|---------|------|------|------|------|------|------|------|------|------|------|
| Samples | 20 | 23 | 38 | 91 | 136 | 215 | 226 | 499 | 535 | 601 |

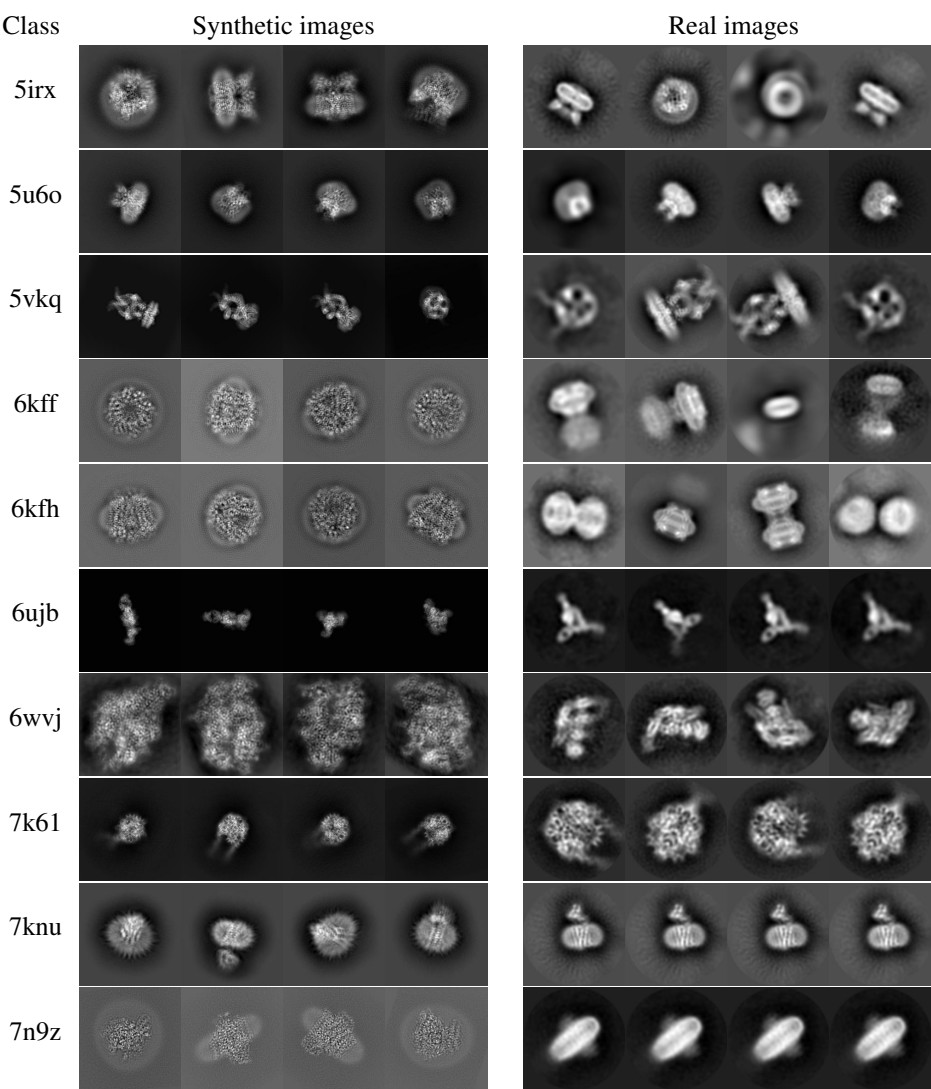

Figure 1: Visualization of the ten selected classes from the synthetic and real datasets. Some classes appear similar in both datasets (5irx, 5u6o), but there are differences in scale (5vkq), limited diversity (7knu, 7n9z), and lipid membrane appearance (white ring visible in, for example, 6kff and 6kfh)

## 4 METHOD

In this section, we describe our proposed approach, which consists of a two-stage training paradigm for learning robust retrieval embeddings from cryo-EM 2D class averages, together with an augmentation pipeline applied during training.

**Two-stage training paradigm** We adopt a two-stage training paradigm motivated by both the limited availability of labeled experimental cryo-EM data and the need to retrieve real 2D class averages using synthetic references. The overall goal is to learn an embedding space in which

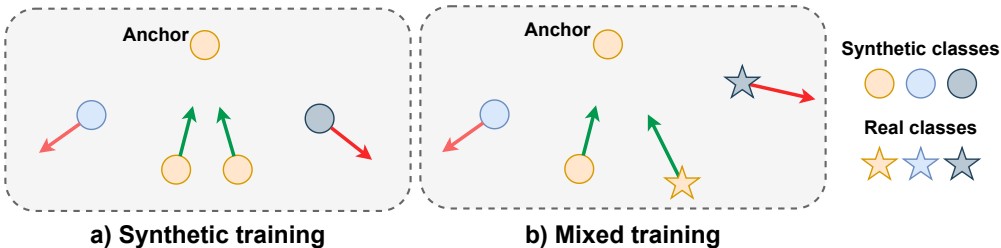

Figure 2: Diagram of the proposed two-stage training approach. A triplet loss is used to encourage embeddings of samples from the same class to cluster. (a) The model is first pretrained on synthetic data (circles). (b) It is then finetuned on a balanced mix of synthetic (circles) and real data (stars) to align embeddings across domains.

synthetic and experimental samples are directly comparable. An overview of the training process is shown in Figure 2.

**Stage I: Synthetic pretraining**   In the first stage, the model is trained exclusively on synthetic 2D class averages. Real experimental data is scarce and often unevenly distributed across classes, making it insufficient for learning a robust embedding space from scratch. In contrast, synthetic data can be generated in large quantities with controlled variability, enabling the network to learn stable, discriminative feature representations and to establish synthetic class averages as reliable anchors in the embedding space.

**Stage II: Mixed finetuning with real data**   In the second stage, the model is finetuned using a mixture of synthetic and real 2D class averages. This stage is designed to enable retrieval of experimental samples using synthetic anchors by explicitly aligning both domains in a shared embedding space. To prevent bias toward either domain, synthetic and real samples are combined in a balanced manner on a per-class basis. Mixed finetuning encourages experimental samples to cluster around their corresponding synthetic counterparts, improving cross-domain retrieval while preserving class-level discrimination. An additional benefit of this two-stage design is its computational efficiency: finetuning a pretrained model on a small mixed dataset is substantially cheaper than training a large model from scratch. This makes the approach practical for scaling to larger backbones and for iterative updates as new experimental data become available.

**Training**   To train the retrieval model, we employ a triplet margin loss combined with a mining strategy that selects all triplets violating the margin constraint. This encourages embeddings of similar samples to be pulled closer together while pushing apart those of dissimilar samples, thereby improving the discriminative power of the learned representation.

**Augmentations**   We propose a new augmentation pipeline tailored to cryo-EM 2D class averages. Random resized cropping is applied to introduce scale invariance. Random affine transformations, including rotations of up to $180°$ and translations of up to $10\%$ of the image size, enforce rotational invariance and account for off-centered class averages. Random histogram shifts simulate variations in imaging parameters and microscope conditions. Pixel dropout with a probability of $0.2$ is used to mitigate overfitting.

To achieve resolution invariance while preserving characteristic structural features, we apply low-pass filtering with cutoff frequencies sampled uniformly from $[0.05, 1]$. Finally, a soft circular mask with a cosine falloff is applied to replicate the masking patterns commonly observed in real 2D class averages.

## 5 EXPERIMENTS

### 5.1 EXPERIMENTAL SETUP

All experiments use a Vision Transformer (ViT) backbone Dosovitskiy et al. (2020), specifically the `vit_base_patch16_224` variant (85M parameters) from the `timm` library. The input projection layer is adapted to accept single-channel images, and the classification head is removed to obtain a feature extractor suitable for retrieval. All models are trained using mixed-precision and optimized with AdamW Loshchilov & Hutter (2019). The learning rate is initialized at $1 \times 10^{-4}$ and decayed following a cosine annealing schedule to a final value of $1 \times 10^{-6}$.

Model selection is performed based on $k$-nearest neighbor retrieval accuracy on the validation split, consistent with the retrieval objective. All experiments are conducted on a single NVIDIA A100 GPU.

### 5.2 TRAINING REGIMES

**ImageNet pretrained baseline** To assess the transferability of generic visual representations to cryo-EM retrieval, we evaluate the ViT backbone pretrained on ImageNet. The model is initialized with ImageNet pretrained weights, adapted to single-channel input, and used as a fixed feature extractor without further finetuning.

This setting provides a reference point for the benefits of domain-specific pretraining and synthetic anchoring. Notably, to the best of our knowledge, there are no publicly available pretrained foundation models specifically tailored to cryo-EM 2D class averages. As a result, ImageNet pretraining represents the most practical initialization for comparison, despite the substantial domain gap between natural images and cryo-EM data.

**Synthetic-only training** To establish synthetic embeddings and evaluate performance in the absence of real data, the model is trained exclusively on synthetic 2D class averages for 20,000 steps using a batch size of 1024. Validation is performed every 1,000 steps. The synthetic training dataset contains 175 classes (see Section 3.1), with 20 reference samples per class.

**Real-only training** To assess performance when training solely on experimental data, the model is trained on real 2D class averages for 5,000 steps using a batch size of 256, with validation every 100 steps. The dataset contains 10 real classes (see Section 3.2), with 5 reference samples per class. This setting reflects the limited scale of available experimental data.

**Mixed training** As a baseline for joint domain learning without pretraining, the model is trained from scratch on a mixed dataset of paired synthetic and real samples for 5,000 steps using a batch size of 256, with validation every 100 steps. The dataset consists of 10 classes, with synthetic and real samples balanced per class and per domain, and 5 reference samples per class.

**Synthetic pretraining followed by mixed finetuning** To implement the proposed two-stage training paradigm, the model is first pretrained on synthetic data and subsequently finetuned on a mixed synthetic–real dataset. Starting from the synthetic-only training checkpoint, the model is further finetuned for 5,000 steps with a batch size of 256 and validation every 100 steps. The finetuning dataset matches the mixed training setting, consisting of 10 paired synthetic and real classes with balanced sampling and 5 reference samples per class and domain.

### 5.3 EXPERIMENTAL RESULTS

We evaluate all training regimes described in Section 5.2 in a retrieval setting with **synthetic** class averages as anchors and **real** class averages as queries. Retrieval is performed using cosine similarity in the learned embedding space. Performance is measured using recall at one (R@1), recall at five (R@5), and $k$-nearest neighbor accuracy with $k = 5$.

During evaluation, we consider 185 classes represented by five synthetic anchors per class. Queries are drawn from ten real classes, using 20% of the available samples from each class. Quantitative results are reported in Table 2.

Table 2: Performance comparison of different training regimes for cryo-EM 2D class average retrieval. ImageNet pretrained baseline or real-only training perform poorly in the cross-domain retrieval task. Training on synthetic or mixed datasets improves performance, while the best results are achieved with synthetic pretraining followed by mixed finetuning. Metrics reported are recall@N (R@N) and $k$-nearest neighbor accuracy.

| Experiment | R@1 | R@5 | Accuracy |
|---|---|---|---|
| ImageNet pretrained baseline | 0.01 | 0.01 | 0.01 |
| Synthetic-only training | 0.24 | 0.31 | 0.20 |
| Real-only training | 0.12 | 0.36 | 0.13 |
| Mixed training | 0.74 | 0.99 | 0.94 |
| Synthetic pretraining + mixed finetuning | **0.96** | **1.00** | **0.98** |

The **ImageNet pretrained baseline** reports zero performance for all metrics, reflecting a severe domain mismatch between natural images and cryo-EM 2D class averages. This confirms that generic visual features learned from natural image statistics do not transfer to the cryo-EM retrieval setting.

The **synthetic-only training** yields a modest improvement over the ImageNet baseline, indicating that training on domain-relevant synthetic data enables the model to capture coarse structural features. However, performance remains limited due to the absence of real data during training, which prevents effective alignment between synthetic anchors and experimental queries.

The **real-only training** shows similar performance to synthetic-only training, highlighting the difficulty of learning a discriminative and domain-aligned embedding space from a small number of real samples alone. Without synthetic anchors during training, the model fails to bridge the domain gap required for cross-domain retrieval.

The **mixed training**, in which synthetic and real samples are jointly used from scratch, results in a substantial performance gain across all metrics. This demonstrates that explicit exposure to both domains during training encourages alignment in the embedding space and significantly improves synthetic-to-real retrieval.

The best performance is achieved by **synthetic pretraining followed by mixed finetuning**, which attains near-perfect recall and accuracy. Pretraining on abundant synthetic data establishes stable synthetic anchors, while subsequent mixed finetuning aligns real samples to these anchors. These results confirm that, under limited real-data availability, synthetic data provides an effective prior that substantially enhances cross-domain retrieval performance.

# 6  LIMITATIONS

The main limitation of our study is the lack of annotated real cryo-EM 2D class average datasets. While our two-stage domain-mixed paradigm tries to mitigate this issue by leveraging synthetic pretraining, the limited availability of real data constrains both the diversity of experimental conditions represented and the scope of downstream benchmarking. Consequently, retrieval performance may not fully capture the variability encountered in large-scale cryo-EM studies.

# 7  CONCLUSIONS

We presented a retrieval framework for cryo-EM 2D class averages and introduced a two-stage training paradigm based on synthetic pretraining followed by mixed finetuning. This strategy addresses the scarcity and imbalance of experimental data by using synthetic samples to establish stable anchors in the embedding space, enabling effective alignment of real class averages. Across all evaluated settings, the proposed approach substantially outperforms training on real data alone.

Our results demonstrate that synthetic data can serve as a strong and scalable prior for representation learning in cryo-EM, providing a practical mechanism for bridging synthetic and experimental domains. This framework offers a foundation for extending retrieval-based analysis to larger and more diverse cryo-EM datasets, with potential applications in high-throughput structural analysis.

Future work will focus on reducing the remaining domain gap through domain adaptation and generative modeling. In particular, generative AI techniques could be used to produce more realistic and diverse synthetic class averages, further improving cross-domain alignment and retrieval performance.

### MEANINGFULNESS STATEMENT

We consider a meaningful representation of life to be one that captures biologically relevant structure while remaining robust to noise, variability, and incomplete observations inherent to experimental data. In this work, we contribute to this direction by learning embedding spaces for cryo-EM 2D class averages that preserve structural similarity across resolution, orientation, and domain boundaries. By aligning synthetic and experimental representations, our approach enables scalable comparison and retrieval of molecular structures, supporting data-driven organization of biological diversity and facilitating downstream structural and functional analysis.

### THE USE OF LARGE LANGUAGE MODELS

We used LLMs to find better synonyms and correct grammar in our text to improve its readability. We read, analyzed, and modified each LLM suggestion, if needed, to ensure that there were no hallucinations that could lead to misinformation.

### ACKNOWLEDGMENTS

The work conducted by Radosław Kuczbański, Wojciech Kozłowski and Maciej Zięba was supported by the National Centre of Science (Poland) grant no. 2021/43/B/ST6/02853.

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
