# OpenReview forum: "Learning image representation for limited cryo-EM data"
_ICLR.cc/2026/Workshop/LMRL — ICLR 2026 Workshop LMRL Poster_

### Official Review · Reviewer_HSPE · 2026-02-25
**Good results but lack of motivation and clarity**

**Rating:** 5
**Confidence:** 2

**Review:**

This work introduces a cross-domain image retrieval task for cryo-EM 2d class averages, and proposes to solve it with a two-step learning paradigm combining synthetic data in the pretraining step and mixed (real and synthetic) data in the finetuning step.

Strength: The proposed synthetic pretraining + mixed finetuning method is highly effective and obtains largely better performance than other baselines considered (Table 2).

Lack of motivation and clarity: In general, the motivation of the submission is not clear. More details are needed in the explanation of 2d class averages (line 069), in particular what are the inputs and outputs of methods to compute 2d class averages, or what causes differences between images of a given class, for instance 5irx in figure 1 (I assume the orientation?). Similarly, one of the two contributions of the work is the cross-domain image retrieval task, but the submission lacks a clear motivation for the task, an explicit description of it, and an explanation of how it differs from similar tasks used in the field. I am not convinced of the importance of this task, especially compared to a task that evaluates performance on real data only.

Also, the submission neglect to consider a natural baseline: pretraining on synthetic and finetuned on real (instead of mixed) data, which makes it hard to evaluate the benefit of the mixed finetuning, compared to a more obvious approach.

Originality: This work is original because it proposes a novel task, but the originality of the training paradigm itself is low.

Minor comments:
- CTF is not defined (line 68)

---

### Meta-Review · Area_Chair_tNPu · 2026-02-28

**Recommendation:** Accept (Poster)
**Confidence:** 4

**Metareview:**

Nice use of synthetic data to offset the relatively small amounts of real cyroEM data.

---

### Decision · Program_Chairs · 2026-03-02

**Decision:**

Accept (Poster)

**Comment:**

Please see the meta-review.